# A Score to Predict the Malignancy of a Breast Lesion Based on Different Contrast Enhancement Patterns in Contrast-Enhanced Spectral Mammography

**DOI:** 10.3390/cancers14174337

**Published:** 2022-09-05

**Authors:** Luca Nicosia, Anna Carla Bozzini, Simone Palma, Marta Montesano, Filippo Pesapane, Federica Ferrari, Valeria Dominelli, Anna Rotili, Lorenza Meneghetti, Samuele Frassoni, Vincenzo Bagnardi, Claudia Sangalli, Enrico Cassano

**Affiliations:** 1Breast Imaging Division, Radiology Department, IEO European Institute of Oncology IRCCS, 20141 Milan, Italy; 2University Department of Radiological and Hematological Sciences, Catholic University of the Sacred Heart, Largo Francesco Vito 1, 00168 Rome, Italy; 3Department of Statistics and Quantitative Methods, University of Milan-Bicocca, 20126 Milan, Italy; 4Data Management, European Institute of Oncology IRCCS, 20141 Milan, Italy

**Keywords:** contrast-enhanced spectral mammography (CESM), breast carcinoma, breast, biopsy, score

## Abstract

**Simple Summary:**

Contrast-enhanced spectral mammography (CESM) represents a novel, reliable imaging adjunct for the early detection and management of breast lesions by coupling high sensitivity and specificity. Early diagnosis of breast tumors using this innovative diagnostic method could help in reducing the high number of unnecessary biopsies for breast lesions considered as suspicious by standard radiological examination (such as mammography, ultrasound, and magnetic resonance). Considering its relative recent introduction, there are still no standardized methods for assessing the diagnostic performance of CESM enhancement descriptors. Here, we aim to create a score that takes account four CESM enhancement descriptors able to efficiently predict the malignancy of a breast lesion prior to biopsy.

**Abstract:**

Background: To create a predictive score of malignancy of a breast lesion based on the main contrast enhancement features ascertained by contrast-enhanced spectral mammography (CESM). Methods: In this single-centre prospective study, patients with suspicious breast lesions (BIRADS > 3) were enrolled between January 2013 and February 2022. All participants underwent CESM prior to breast biopsy, and eventually surgery. A radiologist with 20 years’ experience in breast imaging evaluated the presence or absence of enhancement and the following enhancement descriptors: intensity, pattern, margin, and ground glass. A score of 0 or 1 was given for each descriptor, depending on whether the enhancement characteristic was predictive of benignity or malignancy (both in situ and invasive). Then, an overall enhancement score ranging from 0 to 4 was obtained. The histological results were considered the gold standard in the evaluation of the relationship between enhancement patterns and malignancy. Results: A total of 321 women (median age: 51 years; range: 22–83) with 377 suspicious breast lesions were evaluated. Two hundred forty-nine lesions (66%) have malignant histological results (217 invasive and 32 in situ). Considering an overall enhancement score ≥ 2 as predictive of malignancy, we obtain an overall sensitivity of 92.4%; specificity of 89.8%; positive predictive value of 94.7%; and negative predictive value of 85.8%. Conclusions: Our proposed predictive score on the enhancement descriptors of CESM to predict the malignancy of a breast lesion shows excellent results and can help in early breast cancer diagnosis and in avoiding unnecessary biopsies.

## 1. Introduction

### 1.1. Breast Cancer and Radiologist Role

Breast cancer (BC) is the most common cancer worldwide, and one of the most common causes of death in women; in 2020, there were 2.3 million women with breast cancer and 685,000 deaths globally [1]. Early and effective diagnosis has always been the challenge of breast radiology, aimed at reducing mortality and offering better management of those patients. The main goal in breast imaging is the early detection of neoplasms and avoidance of unnecessary biopsies [2,3]. Contrast-enhanced spectral mammography (CESM) seems to be a promising method in this regard [4].

### 1.2. Current Breast Imaging Methods

Nowadays, BC diagnosis and staging are based on three main diagnostic techniques: full-field digital mammography (FFDM), ultrasound (US), and magnetic resonance imaging (MRI). Currently, FFDM is still the method of choice for breast cancer screening [5], although high breast tissue density among women can reduce the sensitivity [6,7,8]. Consequently, advanced mammography techniques, such as CESM, were developed to overcome FFDM limitations. Compared with FFDM, by administering contrast medium and creating subtracted images from high- and low-energy acquisitions, CESM improved cancer detection and decreased the misdiagnosis rate, especially in dense breasts [9,10,11,12].

### 1.3. Limitations of Imaging Methods: The Role of a Predictive Score of CESM Enhancement

Women with dense breasts are approximately 40% of patients undergoing breast screening [13]. In these cases, FFDM has lower sensitivity, with a high rate of false-negative results. On the other hand, US is particularly suitable for studying dense breasts, but its overall diagnostic efficacy is affected by interpretative issues: a restricted field of view and high operator dependency [14,15]. Furthermore, the difficulty of US (compared to mammography) to depict microcalcifications reduces the sensitivity in those forms of breast malignancies that present only as microcalcifications [16,17,18]. Finally, MRI is characterized by a higher sensitivity than FFDM and US for its ability to detect tumor angiogenesis through contrast enhancement [19,20,21,22]. However, the specificity of MRI is worsened by a high false-positive rate [23,24,25]. Other limitations of MRI include high equipment and examination costs, and it is not possible to perform MRI on all patients (e.g., pacemaker, metallic devices) [26,27,28]. All main diagnostic techniques for the detection of breast malignancies have some flaws: FFDM has low sensitivity in dense breast [6,7,8], US is too operator-dependent and has a restricted field of view [14,15], and MRI has many false positives, and a high cost and execution time [23,24,25].

In this scenario, it is now well-known that CESM could represent a reliable imaging adjunct [29,30,31,32] for the early detection of BC, by coupling sensitivity to specificity, thus, representing a possible solution to the weaknesses of the main diagnostic methods. However, it is still poorly understood what and how many contrastographic enhancement features (and their possible combinations) of a breast lesion occur during CESM, and how they can be used to effectively predict the malignancy of a lesion [33,34,35,36].

The aim of our prospective work is to evaluate the diagnostic performance of each known enhancement descriptor of CESM, and to create a predictive score of breast lesion malignancy based on their combination. We evaluate the sensitivity, specificity, positive and negative predictive values, and diagnostic accuracy of our score.

## 2. Materials and Methods

The study was conducted according to the guidelines of the Declaration of Helsinki and approved by the Ethics Committees of the European Institute of Oncology (Protocol Number IEO S626/311 and IEO 960; EUDRACT Number 2019-000326-22; approved date 30 March 2012 and 7 September 2020). All the patients signed a specific informed consent form.

### 2.1. Study Design and Population

We enrolled all the patients with at least one dubious lesion of the breast (BIRADS > 3) at the conventional screening examinations with FFDM and/or US performed in our institution. All the patients in this study underwent CESM prior to cito/histological assessment in a period between January 2013 and February 2022. All the lesions were classified according to the Breast Imaging Reporting and Data System (BI-RADS) [37]. Breast density and background parenchymal enhancement were also recorded [37]. A radiologist with more than 20 years’ experience in breast imaging evaluated the presence or absence of enhancement, and defined the following enhancement descriptors for each lesion:Intensity of enhancement: absent, mild, moderate, and marked. Absent and mild were considered predictive of benignity, moderate and marked predictive of malignancy;Margin morphology: absent, regular, and irregular. Absent and regular were considered predictive of benignity, irregular predictive of malignancy;Pattern: absent, homogeneous, heterogeneous, and ring. Absent and homogeneous were considered predictive of benignity, heterogeneous, and ring predictive of malignancy;Ground glass: absent, purified, and unpurified. Absent and purified were considered predictive of benignity, unpurified predictive of malignancy.

For each descriptor, a score of 0 was given in the case of predictive benign features (e.g., mild for intensity) and 1 in the case of predictive malign features (e.g., irregular for margins). For each lesion, an enhancement score ranging from 0 to 4 was obtained. A score ≥ 2 was considered predictive of malignancy. The biopsy or surgical histological results were considered the gold standard in the evaluation of the relationship between enhancement patterns and malignancy. The B3 lesions were considered as benign [38].

Finally, for each lesion, the morphology of the enhancement was described (absent, non-mass, and mass) and related with the type of lesion (invasive or in situ) to surgery (for malignant lesions). Some of the main enhancement features are described in Figure 1, Figure 2 and Figure 3.

Analysis of the operative pieces of lesions that went to surgery was performed by a pathologist with more than 20 years of experience in the field of breast pathology. Once excised, surgical specimens, were taken as quickly as possible (maximum 20 min) to the pathology laboratory for analysis. In case it was not possible to send the surgical specimen immediately to the laboratory, within 20 min from excision it was placed in 10% buffered formalin. Upon arrival at the laboratory, the pathologist performed an initial macroscopic analysis of the specimen and marked the surgical margins with ink. After that, the specimen was sectioned and examined microscopically with definition of tumor histotype or benign pathology, and degree of differentiation assessment, including size, and vascular invasion in case of malignant pathology. In malignant pathology, immunohistochemistry was also performed for receptor, replicate index, and HER2 status evaluation. At the end, a report was written by the pathologist.

### 2.2. Inclusion Criterion of the Study

-Patients with dubious breast lesions (BIRADS > 3) at the conventional imaging examination (FFDM and US) for which a cito/micro histological assessment is recommended;-Patients capable and willing to comply with study procedures, and having signed and dated the informed consent form;-Patients with breast size compatible with the dimensions of the image detector;-Patients in sufficiently good health to undergo a CESM examination. We used Eastern Cooperative Oncology Group (ECOG) performance status scale [39]. Only patients with ECOG scale of 0 or 1 were enrolled in the study;-Patients with an age > 18 years old.

### 2.3. Exclusion Criterion of the Study

-Patients with risks of adverse effects with iodine contrast agents;-Patients with breast implant(s);-Patients with proved or supposed pregnancy. For all women with childbearing potential, a pregnancy test was performed before the CESM;-Patients with severe (i.e., GFR < 30 mL/min), mild, and moderate renal function were excluded from the study;-Patients with manifest hyperthyroidism;-Patients in therapy with a biguanide such as metformin were excluded from the study, due to possible interactions.

### 2.4. Technique

We performed dual-energy contrast-enhanced spectral mammography (CESM) using Senographe^®^ Essential, full-field digital system (GE Healthcare, Chalfont St. Giles, UK), or the Amulet^®^ Innovality^®^ (Fujifilm, Akasaka Minato-ku Tokyo, Japan) or Selenia^®^ Dimension^®^ (Hologic, Marlborough, MA, USA) for dual-energy CESM acquisitions.

Using a power inflator, we injected (through a catheter placed in the arm) 1.5 mL/Kg of contrast agent (Visipaque^®^ 320, GE Healthcare, Chalfont St. Giles, UK) for each patient included in the study. Craniocaudal (CC) and mediolateral oblique (MLO) projections were obtained after two minutes of contrast agent administration with 2 different energy exposures (high and low).

Low-energy acquisitions were acquired with kilovoltage values ranging from 26 to 31 kVp. while high-energy images were acquired at 45–49 kVp.

Then a logarithmic subtraction of the two images was obtained generating contrast-enhanced images with the subtraction of the surrounding parenchyma.

### 2.5. Statistical Analysis

Continuous data were reported as median and ranges. Categorical data were reported as counts and percentages.

The associations between CESM enhancements were measured using the phi coefficient.

The accuracy in defining the malignant or benign nature of the lesions, according to the enhancements, was evaluated using the histological result as the gold standard.

An enhancement score was created, with a value equal to the number of enhancements indicating a malignant lesion. The score is equal to 0 if all the enhancements are predictive of benignancy, while it is4 if all the enhancements are predictive of malignancy.

The ROC curve of the model with the score as independent variable and the histological result as dependent variable was performed, and the cut-off maximizing the Youden index was selected as the optimal. On this basis, a score value equal or greater than 2 was considered as predictive of malignancy.

Sensitivity (SE), specificity (SP), positive predictive value (PPV), negative predictive value (NPV), and diagnostic accuracy (DA), with their 95% confidence interval (95% CI), were calculated for each enhancement and the score enhancement.

Chi-square test and multinomial logistic regression were used to evaluate the enhancement score ability to detect an in situ or an invasive lesion, and to evaluate the association between the morphology (mass vs. non-mass) and invasive lesions.

A *p*-value less than 0.05 was considered statistically significant.

All analyses were performed with the statistical software SAS 9.4 (SAS Institute, Cary, NC, USA).

## 3. Results

We enrolled 321 patients with 377 breast lesions. The median age at the exam is 51 years (22–83). The majority (86.2%) of the examinations are performed with the GE mammograph Senographe^®^ Essential, (GE Healthcare, Chalfont St. Giles, UK). Most of the lesions present as masses (66.0%); less frequently as microcalcifications (26.8%). In 31.3% of cases, lesion have a BIRADS of 4a; in 21.9%, a of BIRADS 4b; in 29.7%, a BIRADS of 4c, and in 17.1%, a BIRADS of 5. In 65.5% of cases, patients have a breast density of ACR C, and in 21.0% of cases an ACR density of B. The background parenchymal enhancement is minimal in the majority of cases (66.3%). The descriptive variables of patients and lesions are summarized in Table 1.

Two hundred forty-nine lesions (66.0%) are identified as malignant by histology: 217 (57.6%) invasive and 32 (8.5%) in situ. The histological results of the surgery and biopsies are fully described in Appendix A.

The enhancement descriptors provided by the radiologist (with more than 20 years of experience in breast imaging) are presented in Table 2. We see enhancement in 274 lesions (72.7% of the total). In summary, regarding the intensity, the most common type of enhancement is marked (28.9%). Regarding the margin morphology, the most common type is irregular (61.8%). Regarding pattern, the most common type is heterogeneous (56.0%). Finally, the most common type of ground glass is unpurified (56.8%).

Intensity, margin morphology, pattern, and ground glass are predictive of malignancy for 55.2%, 61.8%, 56.8%, and 56.8% of cases, respectively. See Table 2.

The distribution of each individual enhancement’s descriptor, and of the enhancement score among levels of histological results, are described in Appendix A.

In Appendix A, the correlation matrix showing the associations between CESM enhancements are reported. The phi coefficient among CESM enhancements ranges from 0.62 between margin morphology and intensity to 0.94 between ground glass and intensity.

The ROC curve of the model with score enhancements as independent variables and histological result (benign vs. any malignant) as a dependent variable is show in Appendix A.

The sensitivity (SE), specificity (SP), positive predictive value (PPV), negative predictive value (NPV), and diagnostic accuracy (DA) of the enhancement score are 92.4% (95% CI: 89.1–95.7%); 89.8% (95% CI: 84.6–95.1%); 94.7% (95% CI: 91.8–97.5%); 85.8% (95% CI: 79.9–91.7%); and 91.5% (95% CI: 88.7–94.3%), respectively. (See Table 3).

The performance of our score is significantly superior in predicting the malignancy of a breast lesion for the invasive neoplasms (excluding the in situ lesions at histological result from the analysis, SE [95% CI] = 94.9% [92.0–97.9%] and PPV [95% CI] = 84.8% [80.3–89.3%]) compared with the in situ (excluding the invasive lesions at histological result from the analysis, SE [95% CI] = 75.0% [60.0–90.0%], *p*< 0.001, and PPV [95% CI] = 9.9% [6.1–13.6%], *p* < 0.001).

Considering the invasive lesion as the event for the histological result and “mass” as the event for morphology, SE (95% CI) = 82.5% (77.4–87.6%) and PPV (95% CI) = 80.3% (75.1–85.5%); while considering “non-mass” as the event for morphology, SE (95% CI) = 14.7% (10.0–19.5%), *p* < 0.001, and PPV (95% CI) = 62.7% (49.5–76.0%), *p* = 0.007.

## 4. Discussion

Effective detection of breast cancer has always been the most important challenge of breast imaging. The main objective of breast radiologists is to detect neoplasms at the earliest stage and to avoid unnecessary biopsies for benign lesions; CESM seems to be a valuable ally in achieving this goal. Conventional diagnostic methods present some flaws: traditional mammography (FFDM) loses accuracy and sensitivity in dense breasts, ultrasound (US) is extremely dependent on the operator, and MRI is expensive, time-consuming, and provides a high number of false positives [40,41,42].

According to one of the most recent metanalysis, published by Xiang et al. [40], including some of the most important published work [10,43,44,45], the diagnostic sensitivity of the CESM is high and comparable to that of MRI. Less encouraging results are obtained with specificity, which seems lower and still comparable with that of MRI, with average values around 0.66 and 0.52.

However, in our opinion, considering the relative novelty of the CESM examination (first approved by the Food and Drug Administration in 2011) [40], it seems that there is a lack of uniform standardization of how to interpret the enhancement’s descriptor.

With our study, we tried to create a score that standardizes and includes most of the enhancement descriptors, in order to achieve an encouraging performance. With our score, and in our experience, the specificity values are also excellent (89.8%), thus, reducing the number of false positives.

Furthermore, in our experience, the diagnostic performance of the enhancement features for the identification of in situ breast malignancies is significantly lower than for infiltrating malignancies.

This may be important when evaluating those patients in whom a diagnostic underestimation (those patients who have an in-situ neoplasm as a result of biopsy and are found to have an infiltrating neoplasm at surgery) of the biopsy with results of DCIS is suspected: a particular type of enhancement may lead to suspect an invasive lesion rather than an in situ one [46,47,48].

Finally, in order to avoid the risk of not detecting some in situ neoplasms, it is always important to consider the diagnostic role of the low-energy acquisitions of CESM, which have a diagnostic power similar to a full-field digital mammography. A dubious lesion found in low-energy acquisitions, even in the absence of enhancement, should always be further investigated [49].

In our study we propose a comprehensive, easy-to-use score of the enhancement descriptors of a lesion to predict its malignancy (or benignity). In our experience, the results are encouraging in a consistent number of patients.

The main limitations of the study are its single-centre nature, and the fact that the enhancement features are evaluated by only one radiologist (although one with many years of experience). We propose that our score be evaluated in other studies to confirm its excellent diagnostic performance.

## 5. Conclusions

An easy-to-use predictive score that considers the CESM’s main enhancement descriptors can be used to predict the malignancy of a breast lesion, combining high sensitivity and specificity.

CESM confirms itself as a valuable diagnostic method that can offset some of the flaws of conventional imaging.

## Figures and Tables

**Figure 1 cancers-14-04337-f001:**
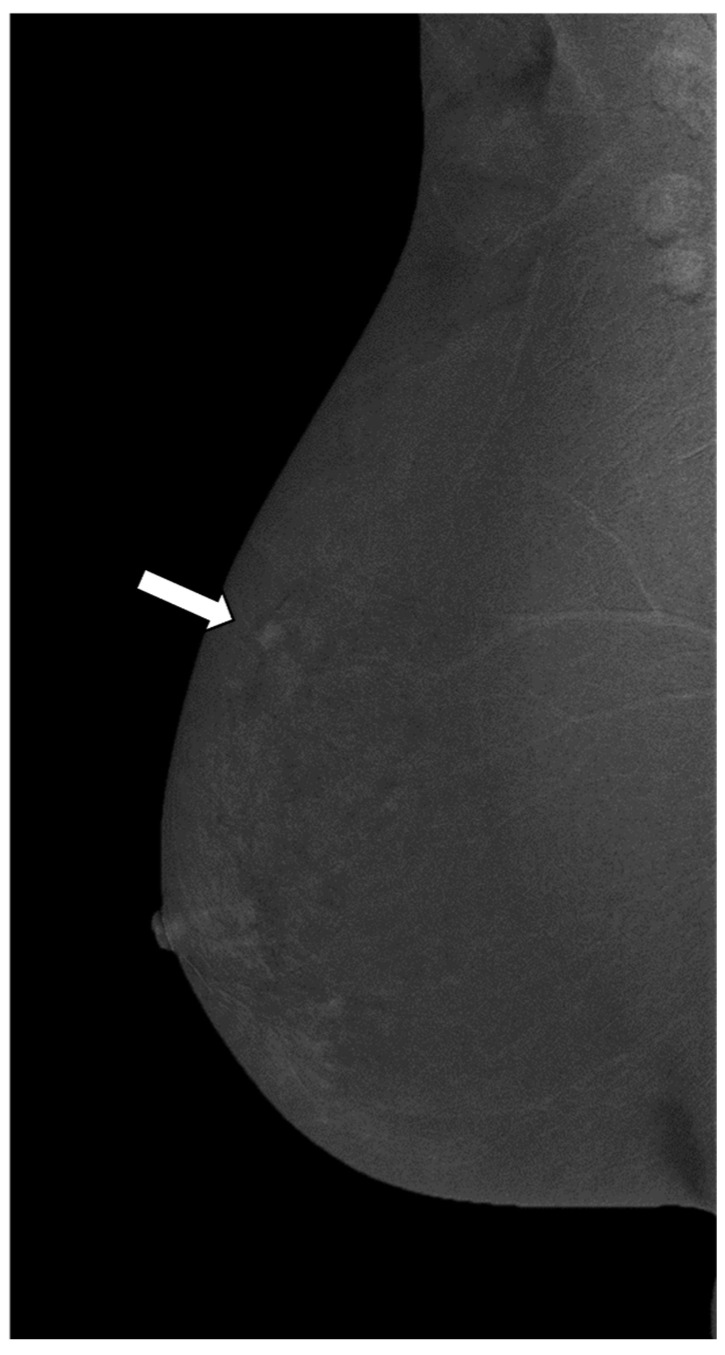
Mediolateral oblique (MLO) view of CESM subtraction images in a 61-year-old woman with mastodynia and dubious ultrasound finding of the right breast (BI RADS 4a). The subtraction image shows a mass enhancement in the upper outer quadrant of right breast (white arrow), measuring less than 10 mm. The enhancement is mild, homogeneous, regular, and purified (enhancement score 0). The biopsy confirms a benign lesion (fibrocystic mastopathy).

**Figure 2 cancers-14-04337-f002:**
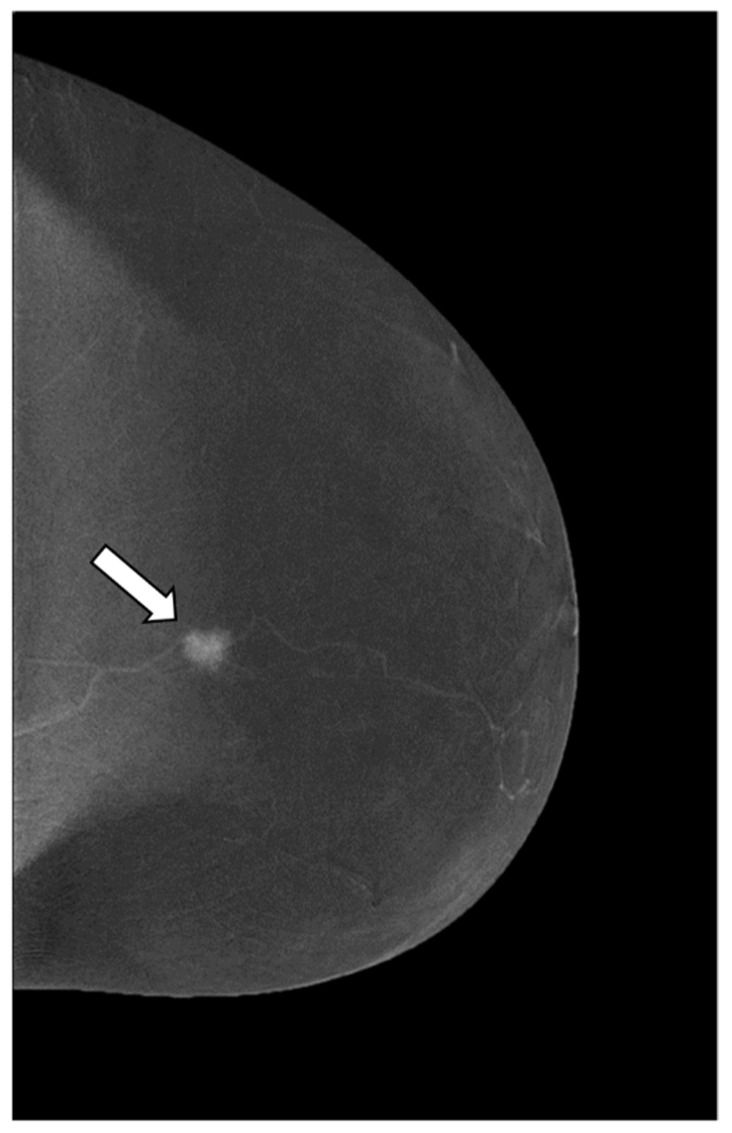
Craniocaudal (CC) view of CESM subtraction image of a 59-year-old patient with a suspicious lesion of the left breast measuring less than 10 mm (BI RADS 4c). The subtraction images show a mass enhancement marked, irregular, heterogeneous, and unpurified (enhancement score 4) (white arrow). The histological result is an invasive ductal carcinoma.

**Figure 3 cancers-14-04337-f003:**
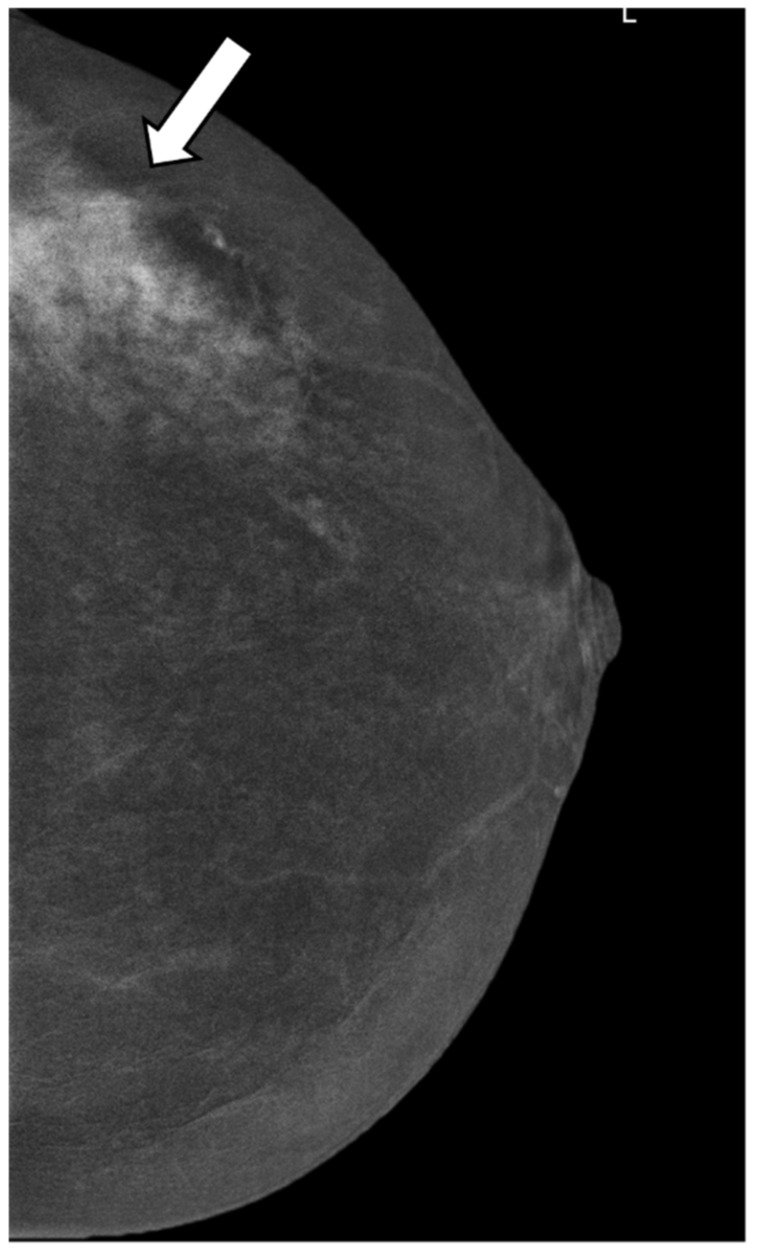
A 41-year-old woman with unilateral hemorrhagic nipple discharge for 1 month. Craniocaudal (CC) view of CESM subtraction images show a non-mass enhancement in the left upper outer quadrant (white arrow) that is moderated, irregular, heterogeneous, and unpurified (enhancement score 4); the histology result shows a high-grade DCIS.

**Table 1 cancers-14-04337-t001:** Descriptive variables (N = 377).

Variable	Level	Overall (N = 377)
Mammograph, N (%)	Fuji	35 (9.3)
	GE	325 (86.2)
	Hologic	17 (4.5)
Type of lesion, N (%)	Microcalcifications	101 (26.8)
	Mass	249 (66.0)
	Mass with microcalcifications	10 (2.7)
	Architectural distortion	8 (2.1)
	Enhancement MRI	5 (1.3)
	Without radiological findings	4 (1.1)
Quadrant, N (%)	Lower	68 (18.0)
	Middle	83 (22.0)
	Upper	226 (59.9)
Side, N (%)	Left	177 (46.9)
	Right	200 (53.1)
BIRADS, N (%)	4a	117 (31.3)
	4b	82 (21.9)
	4c	111 (29.7)
	5	64 (17.1)
	Missing	3
Density (ACR), N (%)	A	4 (1.1)
	B	79 (21.0)
	C	247 (65.5)
	D	47 (12.5)
Background, N (%)	Minimal	250 (66.3)
	Mild	70 (18.6)
	Moderated	35 (9.3)
	Marked	22 (5.8)

**Table 2 cancers-14-04337-t002:** Distribution of the different enhancements and histological result (gold standard) (N = 377).

Variable	Level	Overall (N = 377)
Enhancement, N (%)	No	103 (27.3)
	Yes	274 (72.7)
Intensity, N (%)	Absent	103 (27.3)
	Mild	66 (17.5)
	*Moderate*	99 (26.3)
	*Marked*	109 (28.9)
Intensity (2 categories), N (%)	Benign	169 (44.8)
	*Malignant*	208 (55.2)
Margin morphology, N (%)	Absent	103 (27.3)
	Regular	41 (10.9)
	*Irregular*	233 (61.8)
Margin morphology (2 categories), N (%)	Benign	144 (38.2)
	*Malignant*	233 (61.8)
Pattern, N (%)	Absent	103 (27.3)
	Homogeneous	60 (15.9)
	*Heterogeneous*	211 (56.0)
	*Ring*	3 (0.8)
Pattern (2 categories), N (%)	Benign	163 (43.2)
	*Malignant*	214 (56.8)
Ground glass, N (%)	Absent	103 (27.3)
	Purified	60 (15.9)
	*Unpurified*	214 (56.8)
Ground glass (2 categories), N (%)	Benign	163 (43.2)
	*Malignant*	214 (56.8)
Morphology, N (%)	Absent	103 (27.3)
	Non-mass	51 (13.5)
	Mass	223 (59.2)
Biopsy or surgery histological result, N (%)	Benign	128 (34.0)
	*Malignant* (In situ)	32 (8.5)
	*Malignant (Invasive)*	217 (57.6)
Biopsy or surgery histological result (2 categories), N (%)	Benign	128 (34.0)
	*Malignant*	*249 (66.0)*

**Table 3 cancers-14-04337-t003:** Diagnostic performance of CESM enhancements and the corresponding histological result (benign vs. any malignancy) as the gold standard.

Enhancements	SE [95% CI]	SP [95% CI]	PPV [95% CI]	NPV [95% CI]	DA [95% CI]
Intensity	196/249 = 78.7% [73.6–83.8%]	116/128 = 90.6% [85.6–95.7%]	196/208 = 94.2% [91.1–97.4%]	116/169 = 68.6% [61.6–75.6%]	312/377 = 82.8% [79.0–86.6%]
Margin morphology	220/249 = 88.4% [84.4–92.3%]	115/128 = 89.8% [84.6–95.1%]	220/233 = 94.4% [91.5–97.4%]	115/144 = 79.9% [73.3–86.4%]	335/377 = 88.9% [85.7–92.0%]
Pattern	207/249 = 83.1% [78.5–87.8%]	121/128 = 94.5% [90.6–98.5%]	207/214 = 96.7% [94.4–99.1%]	121/163 = 74.2% [67.5–81.0%]	328/377 = 87.0% [83.6–90.4%]
Ground glass	203/249 = 81.5% [76.7–86.4%]	117/128 = 91.4% [86.6–96.3%]	203/214 = 94.9% [91.9–97.8%]	117/163 = 71.8% [64.9–78.7%]	320/377 = 84.9% [81.3–88.5%]
Enhancement score ≥ 2	230/249 = 92.4% [89.1–95.7%]	115/128 = 89.8% [84.6–95.1%]	230/243 = 94.7% [91.8–97.5%]	115/134 = 85.8% [79.9–91.7%]	345/377 = 91.5% [88.7–94.3%]

SE: sensitivity; SP: specificity; PPV: positive predictive value; NPV: negative predictive value; DA: diagnostic accuracy.

## Data Availability

The data presented in this study are available on request from the corresponding author. The data are not publicly available due to privacy concerns, in accordance with GDPR.

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
