# Peer review of "A Score to Predict the Malignancy of a Breast Lesion Based on Different Contrast Enhancement Patterns in Contrast-Enhanced Spectral Mammography"

_cancers, 2022, doi:10.3390/cancers14174337_

Round 1

Reviewer 1 Report

The ms cancers-1878632 by Nicosia et al tries to develop a predictive score for the first time using the diagnostic performance of CESM. Importantly, they cover the data form

 January 2013 to February 2022 . The sample size is although less i.e.

 321 women and  377 suspicious breast lesions, this study will act as a strong baseline to the field.  The ms is therefore, highly relevant to the field of cancer and be processed further but may be as a rapid communication or short communication.

Therefore few points need to be fixed.

The title is very long. I felt both the parts have similar meaning and therefore be shortened meaningfully by not keeping any abbreviations in it.  

The introduction needs certain modifications. So many small paragraphs need to be fixed.

And most importantly the limitations of all methods such as FFDM, MRI be specified in one paragraphs.

So, the cold be three paragraphs.

1.      About breast cancer and current statistics on it including the 4th paragraph in the present ms.

2.      About all the methods available.

3.      About the limitations of the methods and need of CESM and its predictive score, followed by your hypothesis.

Section 2.4: Although the methodology and statistics are accurate, a specific molecular marker for the confirmation of the state or stage of cancer could be helpful. Also, the evaluation must be performed by the specialists not by only a radiologists.

Please provide the method details such as used in histology.

The limitations of using many molecular markers and the evaluation of the reports by concerned physicists may be done.

Author Response

Reviewer 1

The ms cancers-1878632 by Nicosia et al tries to develop a predictive score for the first time using the diagnostic performance of CESM. Importantly, they cover the data form January 2013 to February 2022 . The sample size is although less i.e. 321 women and  377 suspicious breast lesions, this study will act as a strong baseline to the field.  The ms is therefore, highly relevant to the field of cancer and be processed further but may be as a rapid communication or short communication.

We thank you for your comment and agree that the article stands as a baseline for further future studies. However, because of the size of the sample analyzed and the completeness of the statistical analysis, in accordance with the guidelines for authors of cancers, we believe that the article should belong to the original article category.  We defer to the editor's decision on this specific point.

Therefore, few points need to be fixed.

The title is very long. I felt both the parts have similar meaning and therefore be shortened meaningfully by not keeping any abbreviations in it.  

Thanks for the comment. We changed the title according to your suggestion.

The introduction needs certain modifications. So many small paragraphs need to be fixed.

And most importantly the limitations of all methods such as FFDM, MRI be specified in one paragraphs.

So, the cold be three paragraphs.

  1. About breast cancer and current statistics on it including the 4thparagraph in the present ms.
  2. About all the methods available.
  3. About the limitations of the methods and need of CESM and its predictive score, followed by your hypothesis.

Thank you for your suggestion. We have changed the introduction according to your suggestions.

Section 2.4: Although the methodology and statistics are accurate, a specific molecular marker for the confirmation of the state or stage of cancer could be helpful. Also, the evaluation must be performed by he specialists not by only a radiologist.

Please provide the method details such as used in histology.

The limitations of using many molecular markers and the evaluation of the reports by concerned physicists may be done.

We thank you for the suggestion.

However, the purpose of our work was to create an innovative score based on Contrast Enhancement Spectral Mammography descriptors to predict the malignancy or benignity of a breast lesion.

A molecular analysis was not included for this specific study. We appreciate, however, your suggestion and believe that a molecular analysis can direct future studies on the same topic

Reviewer 2 Report

The paper is well written and methodological correct with interesting conclusions in the proposed field. The paper is methodologically correct.

The research aims to create a predictive score of malignancy of a breast lesion based on the main contrast enhancement features ascertained by contrast-enhanced mammography.

The proposed topic is very interesting and relevant in the field because it could allow to standardize the detection of malignant lesions on contrast-enhanced mammography.

It seems that there is a lack of uniform standardization of how to interpret the enhancement’s descriptor and it could be the first reported  score that can standardize and include most of the enhancement descriptors to achieve an encouraging performance.

The conclusions are consistent with the evidence and they address the main question posed.

The references are appropriate and tables and figures are clear and correct

Author Response

Reviewer 2

The paper is well written and methodological correct with interesting conclusions in the proposed field. The paper is methodologically correct.

 The research aims to create a predictive score of malignancy of a breast lesion based on the main contrast enhancement features ascertained by contrast-enhanced mammography. 

The proposed topic is very interesting and relevant in the field because it could allow to standardize the detection of malignant lesions on contrast-enhanced mammography. 

It seems that there is a lack of uniform standardization of how to interpret the enhancement’s descriptor and it could be the first reported  score that can standardize and include most of the enhancement descriptors to achieve an encouraging performance.

The conclusions are consistent with the evidence and they address the main question posed. 

The references are appropriate and tables and figures are clear and correct

We thank the reviewer for the very favorable comments on our work

Round 2

Reviewer 1 Report

The authors have made some changes but still many of the original comments need to be fixed for for further consideration of the ms. Please remember that the editors or the reviewers have no time to read the same old ms if not modified as per the comments. Following major points (a, b and c) need to be fixed.

a. The Introduction has many small paragraphs.

It should be divided to three paragraphs as follows. So either merge them or rewrite as follows

  1. Paragraph 1: About breast cancer and current statistics on it including the 4thparagraph in the present ms.
  2. Paragraph 2: About all the methods available.
  3. Paragraph 3: About the limitations of the methods and need of CESM and its predictive score, followed by your hypothesis.

b. Section 2.4: The evaluation of the reports must be performed by a specialists, cancer physiologist or person working on mammary gland cancer not by only a radiologist as done in the study.

c. Please provide the method details such as used in histology.

d. Reference section must be enriched with recent references, at least up to 50.

Author Response

  1. A) The Introduction has many small paragraphs.

It should be divided to three paragraphs as follows. So either merge them or rewrite as follows

  1. Paragraph 1: About breast cancer and current statistics on it including the 4thparagraph in the present ms.
  2. Paragraph 2: About all the methods available.
  3. Paragraph 3: About the limitations of the methods and need of CESM and its predictive score, followed by your hypothesis.

Answer: thank you very much for the suggestion, we modified the introduction chapter dividing it in three paragraphs as indicated.

  1. B) Section 2.4: The evaluation of the reports must be performed by a specialists, cancer physiologist or person working on mammary gland cancer not by only a radiologist as done in the study.

Answer: in the material and methods section (section 2 of ms), we have specified that reports were evaluated by pathologist with breast experience. It should also be considered that our work and our score have a specific radiological focus: enhancement features, on which the score was created, could not be evaluated by any other specialist than a radiologist.

C). Please provide the method details such as used in histology.

Answer: in the material and methods section we have described the specific method that were used in histology evaluation

D). Reference section must be enriched with recent references, at least up to 50.

Answer: we enriched our work with other relevant and recent references, up to 50.

Round 3

Reviewer 1 Report

From the Intro part, serial numbers like 1.1, 1.2 may be removed, only a single part with 1. Introduction would be there.